# De novo network analysis reveals autism causal genes and developmental links to co-occurring traits

Catriona J Miller[1], Evgeniia Golovina[1], Joerg S Wicker[2], Jessie C Jacobsen[3,4], Justin M O'Sullivan[1,5,6,7,8]

**Autism is a complex neurodevelopmental condition that manifests in various ways. Autism is often accompanied by other conditions, such as attention-deficit/hyperactivity disorder and schizophrenia, which can complicate diagnosis and management. Although research has investigated the role of specific genes in autism, their relationship with co-occurring traits is not fully understood. To address this, we conducted a two-sample Mendelian randomisation analysis and identified four genes located at the 17q21.31 locus that are putatively causal for autism in fetal cortical tissue (*LINC02210*, *LRRC37A4P*, *RP11-259G18.1*, and *RP11-798G7.6*). *LINC02210* was also identified as putatively causal for autism in adult cortical tissue. By integrating data from expression quantitative trait loci, genes and protein interactions, we identified that the 17q21.31 locus contributes to the intersection between autism and other neurological traits in fetal cortical tissue. We also identified a distinct cluster of co-occurring traits, including cognition and worry, linked to the genetic loci at 3p21.1. Our findings provide insights into the relationship between autism and co-occurring traits, which could be used to develop predictive models for more accurate diagnosis and better clinical management.**

## Introduction

Autism is a group of neurodevelopmental conditions exhibiting persistent social communication deficits, accompanied by repetitive sensory motor behaviours and restricted interests (American Psychiatric Association, 2013). Researchers have long associated autism with other conditions. For example, Simonoff et al (2008) found that 70% of individuals in a stratified subsample of adolescents with autism were found to have at least one co-occurring condition. Common co-occurring conditions reported with autism include attention-deficit/hyperactivity disorder, anxiety, depressive disorders, and schizophrenia (de Lacy & King, 2013; Croen et al, 2015; Lai et al, 2019; Neumeyer et al, 2019). Thus, it is suggested that autism shares common biological mechanisms with co-occurring phenotypes.

GWAS studies have identified autism-associated SNPs, including those associated with its co-occurring traits (Autism Spectrum Disorders Working Group of The Psychiatric Genomics Consortium, 2017; Pain et al, 2019; Sun et al, 2019). Previous work by our group has investigated the impact of autism-associated SNPs on the biological pathways underpinning autism (Golovina et al, 2021). However, how they relate to the biological intersection between autism and co-occurring traits, or indeed autism risk itself, is still unclear. Most of the autism-associated SNPs are located within non-coding regions of the genome, consistent with the hypothesis that they mark regulatory regions associated with changes in gene expression (Turner et al, 2016; Yuen et al, 2016). These regions, known as expression quantitative trait loci (eQTLs) spatially interact with their target genes, forming regulatory connections which are either cis- or trans-acting (occurring within < 1 Mb, or > 1 Mb, respectively) intrachromosomal or trans-acting interchromosomal. As regulatory interactions are tissue-specific and autism is primarily a neurodevelopmental condition, analysing both fetal and adult brain-specific eQTL information may aid our understanding of the mechanisms through which genetic variants act to increase an individual's chance of developing autism and its co-occurring traits.

To investigate the interconnectivity of complex polygenic traits, network-based analyses using human-derived datasets have been previously used by our group (Golovina et al, 2023). In this study, we identified causative and pleiotropic genes that are affected by SNPs associated with both autism and other traits within human cortical gene regulatory networks (GRNs) at two different developmental stages. Combining our network analysis with pathway analysis, deep learning of regulatory mechanisms, and finally, comparisons with healthcare data, we identified different clusters of genes and traits (e.g., 17q21.31 linked to neurological traits and 3p21.1 linked to cognition and worry) as being associated with autism and its co-occurring traits.

---

[1]The Liggins Institute, The University of Auckland, Auckland, New Zealand   [2]School of Computer Science, University of Auckland, Auckland, New Zealand   [3]School of Biological Sciences, The University of Auckland, Auckland, New Zealand   [4]Centre for Brain Research, The University of Auckland, Auckland, New Zealand   [5]The Maurice Wilkins Centre, The University of Auckland, Auckland, Zealand   [6]Garvan Institute of Medical Research, Sydney, Australia   [7]MRC Lifecourse Epidemiology Unit, University of Southampton, Southampton, UK   [8]Singapore Institute for Clinical Sciences, Agency for Science Technology and Research, Singapore, Singapore

Correspondence: justin.osullivan@auckland.ac.nz

# Results

## Spatially constrained cortical GRN were generated for the fetus and adult

Adult and fetal cortical tissues GRNs were produced using the CoDeS3D pipeline (Fadason et al, 2018) to analyse common SNPs (minor allele frequency ≥ 0.05) to identify spatially constrained eQTLs within adult (i.e., GTEx [Genotype-Tissue Expression] eQTL database [GTEx Consortium, 2020]) and fetal cortical eQTL datasets (Walker et al, 2019) (Fig 1). The resulting GRNs were comprised of 580,032 and 1,050,154 spatial eQTLs for the fetal (Miller, 2023b) and adult (Miller, 2023a) cortical tissues, respectively.

## Two-sample Mendelian randomisation identifies four potential causal autism genes within adult and fetal cortical tissue GRNs

A two-sample Mendelian randomisation (2SMR) study was undertaken using the TwoSampleMR R package (https://github.com/MRCIEU/TwoSampleMR/, version 0.5.6) (Hemani et al, 2018) to identify spatially regulated genes that were putatively causal for autism within the adult and fetal cortical tissue GRNs (Fig S1A). The iPSYCH-PGS 2017 ASD GWAS (Grove et al, 2019) was used as the outcome data. After 2SMR, four genes (LINC02210, LRRC37A4P, RP11-259G18.1, and RP11-798G7.6) were identified as being statistically significant (Bonferroni-adjusted P-value < 0.05) and having a putatively causal role in autism (here on referred to as causal) within the fetal cortical tissue (Fig S1B). These four genes are expressed in the fetal cortical tissue GRN with gene expression values of 17.38 (LINC02210), 8.99 (LRRC37A4P), 0.88 (RP11-259G18.1), and 1.35 (RP11-798G7.6) TPM. Within the adult cortical tissue GRN, only LINC02210 was identified as being statistically significant. It is expressed in the adult cortical tissue GRN (3.97 TPM).

## Fetal and adult protein–protein interaction networks (PPINs) identify traits that are pleiotropic with autism

The fetal and adult cortical GRNs (Fig 1A) were queried with autism-associated SNPs (GWAS Catalog [https://www.ebi.ac.uk/gwas/], $P ≤ 5 × 10^{-8}$; Table S1) and those within LD ($r^2 = 0.8$, width = 5,000 bp; Table S2) to identify eQTL-gene associations and create autism-specific GRNs (Fig 1B). The adult autism-specific GRN consists of 888 cis-acting, 63 trans-acting intrachromosomal, and five trans-acting interchromosomal eQTL-gene pairings. The fetal network contains 1,155 cis-acting, 26 trans-acting intrachromosomal and four trans-acting interchromosomal eQTL-gene regulatory connections.

The STRING database was used to create PPINs that extended the autism specific GRNs by four levels, where the proteins on each level interact with a protein encoded by a gene on the previous level (Fig 1B).

Querying the GWAS Catalog with the eQTLs that were associated with genes encoding proteins from each level of the network identified traits that are associated with autism (Figs 2A, S2, and S3; Table S3). Because of the developmental nature of autism (American Psychiatric Association, 2013), we hypothesized that there would be a mixture of shared and unique traits identified in the fetal and adult

PPINs. Across the five levels (level 0–level 4), 44 traits were shared (52% of the adult network and 46% of the fetal network) (Fig 2B). A higher proportion of the index level traits (level 0) was shared between the fetal and adult PPIN than the outer level traits ($P < 0.0001$). There were 63 significant (hypergeometric test $P < 0.05$) traits on level 0 of the adult PPIN and 57 on level 0 of the fetal PPIN, 39 of which appeared in both (Fig 2C). The finding that most of the shared traits were on the index level of the PPIN is notable. The eQTLs for these level-0 traits target genes whose transcript levels also correlate with autism-associated eQTLs. Thus, these level-0 genes are pleiotropic with autism.

Of the index-level traits identified in the fetal PPIN, 54% were brain- or mood-related (Fig 3). Many of the eQTLs that were responsible for the associations correlated with the transcript levels of genes located at chromosome 17q21.31, including LINC02210 and RP11-259G18.1 which were identified as being causal for autism. Notably, of the genes located in 17q21.31, only KANSL1 was represented on the index level of the adult PPIN (Fig 4). Other clusters of genes encoding proteins present within the fetal PPIN included (Table S4) (1) metabolic traits associated with FADS1/2; (2) traits linked with cognition and worry that associated with two gene clusters (TMEM110, GNL3, STAB1 and ITIH4, NEK4, NT5DC2) located at chromosome 3p21.31; and (3) multiple genes across different locations associated with schizophrenia (Fig 3). Although proteins encoded by other genes are present, these groups provide a clear separation of the main traits that appear on the index level of the fetal PPIN.

The index level of the adult PPIN contained 52% brain- or mood-related traits (Fig 4). Of the non-neurological traits, there were multiple lung-related traits linked to eQTLs associated with CHRNA5 (Table S5). Contrary to the fetal PPIN, there was no detectable clustering of gene loci. However, loci at chromosomes 6p22.1 and 10q24.32 were still associated with autism and co-occurring conditions in the adult cortex. Similar to observations of the fetal PPIN, multiple genes from different chromosomal locations were identified as falling at the intersection between schizophrenia and autism.

## Neurological and non-neurological traits were enriched on outer levels of the adult and fetal PPINs

We identified neurological and non-neurological traits as being enriched for the eQTLs associated with the transcript levels of the genes encoding the interacting proteins present in the outer levels of the PPIN (Figs S4 and S5; Tables S4 and S5). BMI-associated eQTLs were correlated with transcript levels from genes encoding proteins in both the fetal and adult PPINs on level 2 (17 and 57 genes, respectively). Notably, eQTLs on the index and outer levels, of both the fetal and adult PPINs, were associated with Parkinson's disease. Across both PPINs, eQTLs affecting multiple genes at different levels were associated with traits related to plasma fatty acid and serum metabolite levels. Immune-related traits (e.g., white blood cell count) were also associated with eQTLs for genes encoding proteins on the outer levels of both the adult and fetal PPINs.

Pathway enrichment analysis identified 4 and 26 biological pathways in the fetal and adult PPINs, respectively, that were shared between the index and outer protein interaction levels (i.e., level 0 and levels 1–4) (Table S6). Many of these pathways

A

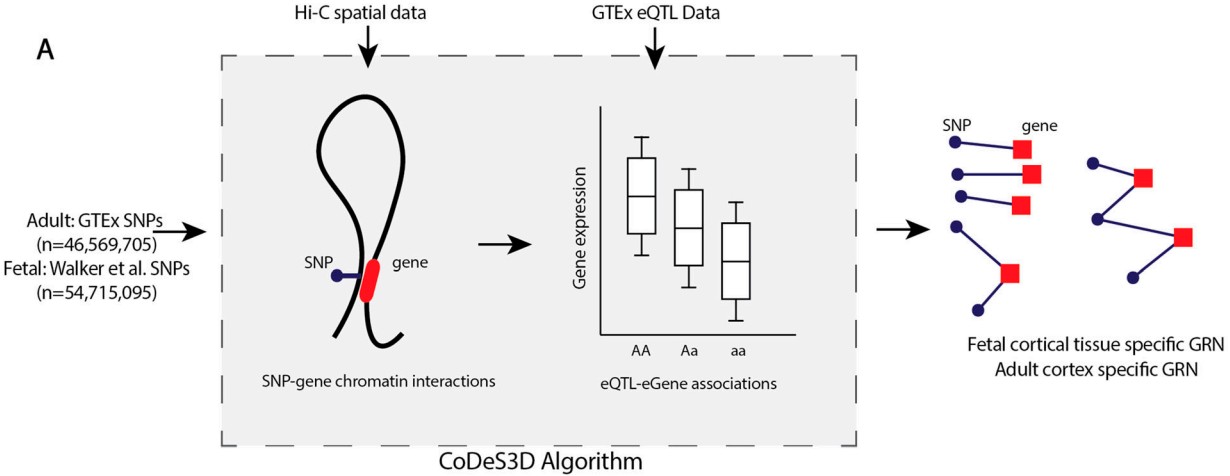

CoDeS3D Algorithm

B

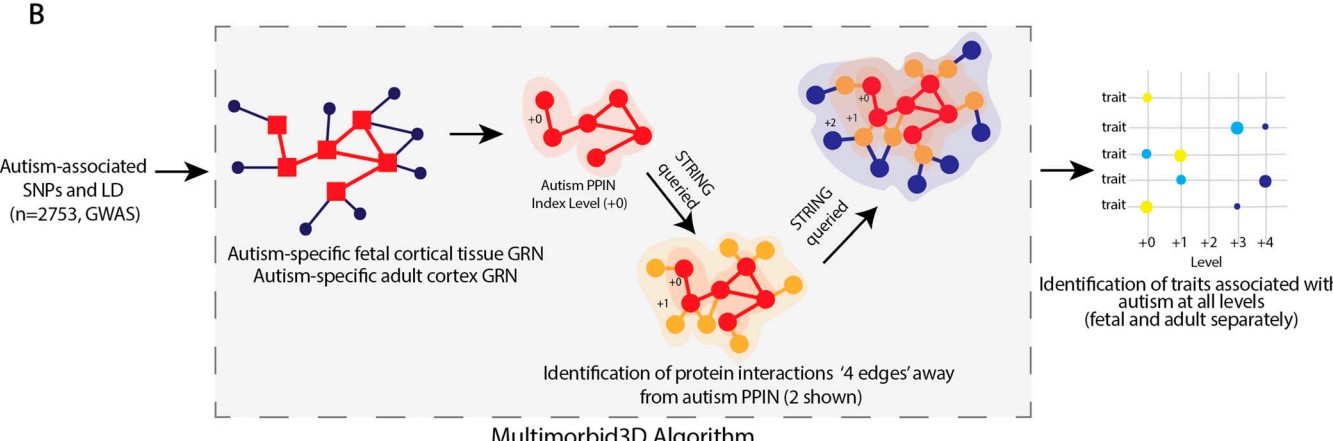

Multimorbid3D Algorithm

C

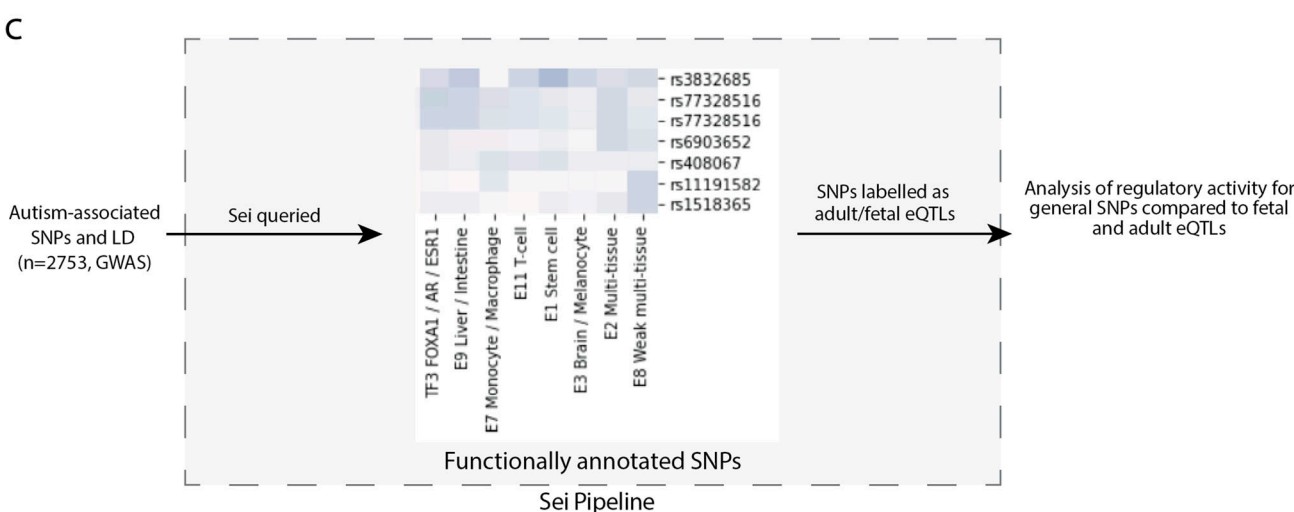

Sei Pipeline

**Figure 1. Schematic of the methods used in this study.**
**(A)** Outline of generation of fetal and adult cortical tissue gene regulatory networks (GRN) using CoDeS3D. Hi-C chromatin libraries, derived from fetal brain-specific cortical plate and germinal zone neurons and adult dorsolateral prefrontal cortex cells, were downloaded from dbGaP (accession: phs001190.v1.p1) and GEO (https://www.ncbi.nlm.nih.gov/geo/, accession: GSE87112) respectively. **(B)** The Multimorbid3D algorithm was used to identify traits that co-occur with autism. Autism-associated

(2 fetal and 14 adult) were disease-related (e.g., asthma, rheumatoid arthritis and type 1 diabetes mellitus), whereas other pathways were immune- or signalling-related (Table S6).

## Deep learning provides insight into regulatory elements associated with autism

The autism-associated SNPs and those in LD (n = 2,753) were queried for patterns of known regulatory elements using Sei (Chen et al, 2022)—a deep-learning algorithm for predicting regulatory activity (Fig 1C). The Sei deep learning sequence model scored each SNP across 40 different regulatory element patterns, 12 of which referred to enhancer activity (Table S7). SNPs were clustered using LD ($r^2$ = 0.8, width = 5,000 bp; Table S2) into 60 fetal and 113 adult eQTL containing loci, and 290 loci containing non-eQTL SNPs (i.e., GWAS SNPs that were not present in the fetal or adult GRNs; Fig S6A). Across these autism-associated genetic loci, Sei predicted that the fetal and adult eQTL containing loci were enriched for enhancers, compared with the 290 loci containing only non-eQTL SNPs (Fig S6B). When looking at the highest scoring regulatory class for each SNP, the fetal and adult eQTLs had a significantly higher mean score than the non-eQTL SNPs (Fig S6C). Notably, there were no significant differences between the adult and fetal eQTL containing loci for enhancer enrichment or mean Sei scores (Fig S6B and C).

The 50 SNPs with the highest Sei regulatory scores (Fig S7) typically scored highest in either (1) the erythroblast-like, multitissue, or weak epithelial enhancer categories; (2) the promoter category or (3) the CTCF–cohesin category. The top 50 SNPs were evenly split between those that were predicted to increase or decrease regulatory activity when compared with the reference sequence at that position. This finding was also true for the top 50 SNPs in the adult and fetal groups (i.e., loci containing at least one eQTL from the adult and fetal GRNs, respectively; Figs S8 and S9). The adult group eQTL (rs308107) that had the highest predicted regulatory score (i.e., 7.93) was associated with *RERE* transcript levels. rs308107 was associated with increases in regulatory activity in enhancer, promotor, and CTCF–cohesin categories (Table S7). rs308107 has been associated with the intersection between major depression (MD) and intelligence, and expression of *RERE* in the brain (Bahrami et al, 2021). *RERE* is a syndromic autism gene (SFARI score S1; https://gene.sfari.org/database/human-gene/RERE) and mutations here have been associated with autism (Fregeau et al, 2016).

## Traits predicted to co-occur with autism are present in autistic New Zealanders

ICD-10 codes from hospital discharge data were used to calculate odds ratios (ORs) for all conditions that co-occur with autism within the population of autistic New Zealanders who had been admitted to hospital between 2015–2020 (n = 2,622; Table S8). Those conditions that were significant (Fisher exact test; $P \leq 0.05$) when compared with the general hospitalised population in the same time-period (n = 2,051,658), after multiple testing correction, were retained and those that overlapped the traits detected by Multimorbid3D were identified (Fig 5). Neurological conditions such as schizophrenia, mood disorders, and depression showed an increased prevalence in autistic New Zealanders (i.e., $\log_{10}(OR) > 0$). These traits were associated with chromosome 17q21.31 in our Multimorbid3D analysis. However, Parkinson's disease was not significantly associated with autism within the New Zealand population. Notably, Alzheimer's disease showed a decreased prevalence in the autistic population. These findings may be because of a smaller number of autistic individuals at an age where Parkinson's disease and Alzheimer's disease generally occur because of recent increases in diagnostic testing (Fombonne, 2009). Alzheimer's disease appeared as a disease pathway in our pathway analysis within the fetal gene group. Although it was not present in the adult group, "pathways of neurodegeneration" was present (Table S6). Abnormal weight gain was associated with over 40 genes on the outer levels of the adult PPIN and had an increased OR in autistic New Zealanders. Chronic obstructive pulmonary disease (COPD) occurred at reduced rates in autistic New Zealanders, and although it was not directly identified in our Multimorbid3D analysis, many lung-related phenotypes including "post bronchodilator FEV1/FVC ratio in COPD" appeared on both the index and outer layers of the adult PPIN (Fig 4).

# Discussion

We performed a de novo network analysis of cortical GRNs that identified four genes that are causal for autism within the 17q21.31 loci in fetal tissue (*LINC02210, LRRC37A4P, RP11-259G18.1, and RP11-798G7.6*), of which, *LINC02210* was also identified as being causally related to autism in adult tissue. Our analysis identified genetic variants that are associated with traits that are known to co-occur with autism (e.g., schizophrenia [Chien et al, 2021; Krieger et al, 2021; Hsu et al, 2022] and BMI [Croen et al, 2015]). We contend that the results of this study provide a starting point for the individualized stratification of the biological mechanisms linking autism and its co-occurring traits.

The nature of the datasets that were used in this work resulted in several limitations (Tam et al, 2019): (1) there is an inherent bias towards common traits that are more "popular" to study by GWAS; and (2) not all genetic variants represented as GWAS participants to date have been primarily of European ancestry. Moreover, the implemented methods assume the proteins encoded by the target genes form characterised protein–protein interactions. Therefore,

---

SNPs ($P < 5 \times 10^{-8}$, n = 576; 2,753 SNPs including those in LD [$r^2$ = 0.8, width = 5,000 bp]) from the GWAS Catalog (www.ebi.ac.uk/gwas; 08/05/2022) were used to query the fetal and adult cortical tissue GRNs to generate the autism-specific GRNs. STRING (version 11.5; https://string-db.org) was queried to create a multilevel protein–protein interaction network for the fetal and adult cortical tissue, separately. **(C)** SNPs were functionally annotated using Sei (Chen et al, 2022). 2,753 SNPs, including those in LD with the autism-associated SNPs (Table S2), were queried into Sei. SNPs were collated into LD loci and separated into three groups (loci containing fetal expression quantitative trait loci [eQTLs], loci containing adult eQTLs, and loci containing only SNPs that were not eQTLs). There was some overlap between the fetal and adult groups.

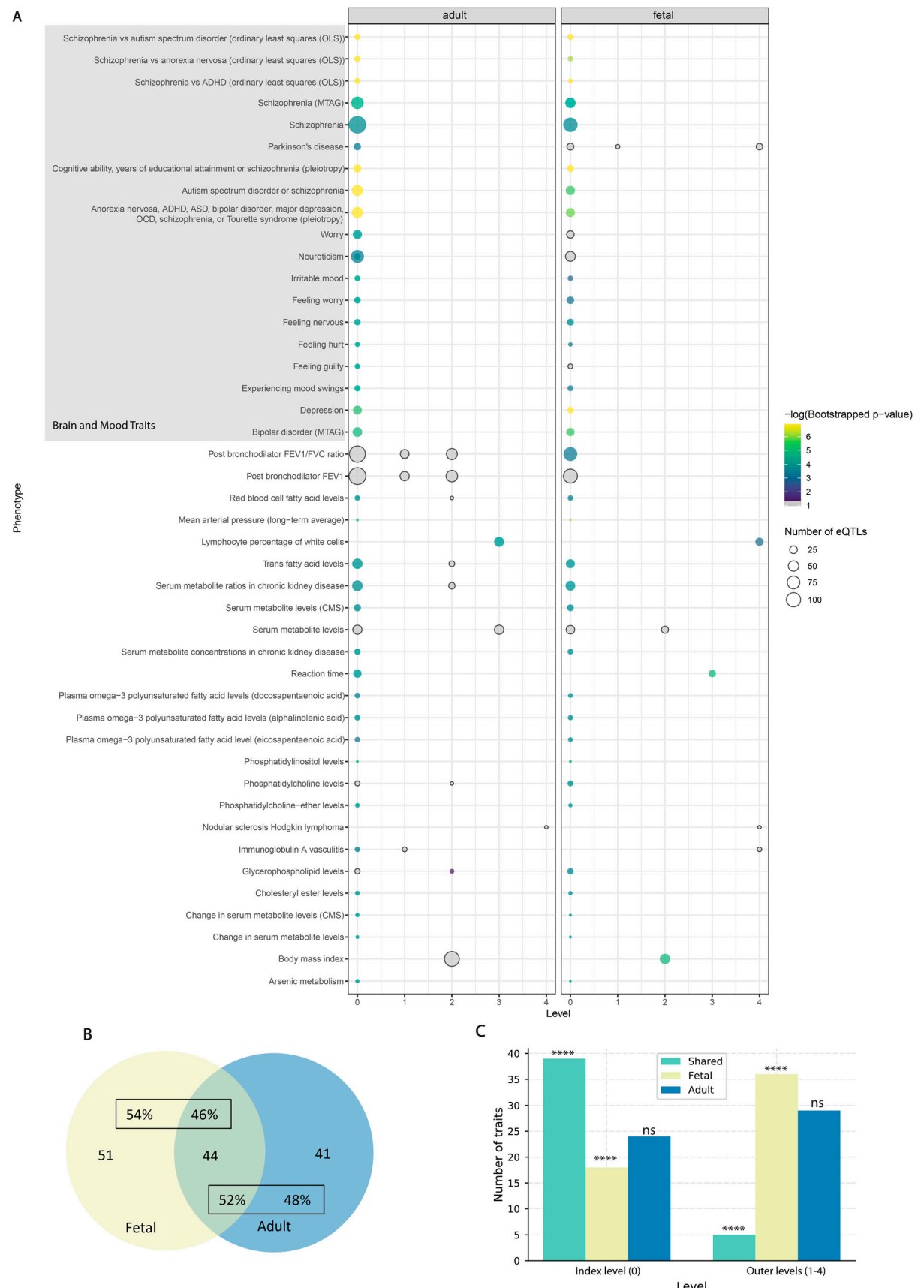

any regulatory connections that associate with genes encoding proteins that do not form, or form unknown, protein–protein interactions will not be identified. Furthermore, transcript levels alone are insufficient to explain protein expression (Battle et al, 2015; Brion et al, 2020; GTEx Consortium, 2020). For example, posttranscriptional processes contribute to expected protein levels (Battle et al, 2015; Brion et al, 2020). Therefore, we must be careful when assuming that there is a direct relationship between changes in eQTL-associated transcript levels, protein expression, and phenotypes. Despite these limitations, the spatially constrained GRNs we have described can be used to identify (1) causal genes, (2) the traits and conditions that co-occur with autism, (3) the genetic risk that links them, and (4) the potential mechanisms responsible for the observed associations. Collectively, these results provide a step change towards understanding how genetic variation contributes to autism and brings us closer to stratification for diagnosis and long-term management according to an individual's combination of co-occurring traits.

The individualistic nature of autism is supported by our findings that link variation in distinct genetic loci (SNPs, genes, and chromosomal regions) to "clusters" of traits. A similar observation was made by Fu et al (2022), who found that a distinct subset of genes contributes to the genetic overlap between autism and developmental disorders, when compared with autism and schizophrenia. Our study also supports this notion, as we found that the overlap between autism and other traits may involve subsets of genes that are specific to the overlap of interest.

Within the fetal PPIN, a common group of genes (i.e., KANSL1, LINC02210, RP11-259G18.3, MAPT-AS1, RP11-259G18.1, and MAPT) were associated with most of the mood/brain-related phenotypes (e.g., neuroticism, Parkinson's disease, depression, mood swings, and white matter microstructure). Many of these phenotypes, particularly depression and mood traits (Croen et al, 2015; Virues-Ortega et al, 2017; Chien et al, 2021), are known to co-occur with autism. Consistent with this, depression and mood disorders were observed to have an increased prevalence in autistic New Zealanders. The genes that associate with these trait intersections are located within chromosome 17q21.31. The 17q21.31 locus is inverted in ~20% of Europeans and has previously been reported as an autism susceptibility locus (Pain et al, 2019). There is also a micro-duplication syndrome at this locus linked to autistic features (Grisart et al, 2009). Notably, we have identified that the regulatory impacts on the genes within this locus involve genes that are causally related to autism and occur in fetal cortical tissue—not the adult cortex. The known functions of the proteins encoded by genes within 17q21.31 provide possible insights into this relationship, albeit not the causal genes which are noncoding or pseudogenes. For example, MAPT encodes tau, potentially resulting in the development of tauopathy in the brain (Grigg et al, 2020). An increase in tauopathy has been identified in postmortem brain sections from an autistic child (Grigg et al, 2020). High MAPT mRNA levels have also been identified in fetal stages of the frontal cortex when compared with similar samples from childhood and adulthood (Xie et al, 2021). KANSL1 has already been identified as having a syndromic causative role in autism with a SFARI score of S1, indicating the gene has been clearly implicated in autism (https://gene.sfari.org/database/human-gene/KANSL1). Koolen-de Vries syndrome, partially characterised by developmental delays and behavioural features, is caused by microdeletions or loss-of-function mutations in KANSL1 (Koolen et al, 2016). With respect to the causally related genes, RP11-259G18.1 has been observed to be up-regulated in the fetal brain (Pain et al, 2019). However, neither LINC02210 nor RP11-259G18.1 have previously been identified as causal autism genes. As the causal genes identified are all noncoding or pseudogenes, little is known about their function. Future work should explore the role these four genes play in the development of autism.

A gene cluster (i.e., NEK4, GNL3, NT5DC2, TMEM110, STAB1, and ITIH4) located at chromosome 3p21.1 was identified as associating with co-occurring traits within the fetal PPIN. Proteins encoded by three genes (i.e., RFT1, ITIH4, and TMEM110) within 3p21.1 were also present within the adult PPIN. Multimorbid3D identified that the genetic variants affecting gene expression within the chromosome 3p21.1 locus are involved in the intersection between autism and cognition, worry, and intelligence. Notably, summary-based Mendelian randomisation and numerous GWAS studies (Scott et al, 2009; Yang et al, 2020; Eum et al, 2021) have associated chromosome 3p21.1 with psychiatric disorders. However, we did not observe a causal relationship within the fetal and adult cortical tissues. Links with autism are largely restricted to case and family studies of NT5DC2, which provide conflicting results on the gene's significance (Wang et al, 2019; Xie et al, 2020). Overexpression of NEK4 and GNL3 in mice has been linked to reduced mushroom spine density in neurons, which has been associated with cognition and memory (Yang et al, 2020).

Although there was a difference in gene clusters and traits affected by spatially constrained eQTLs within the fetal and adult cortical tissues, there were also notable commonalities. For example, Multimorbid3D identified a large genetic intersection between autism and schizophrenia. The connection between autism and schizophrenia has been observed in epidemiological studies (Chien et al, 2021; Krieger et al, 2021; Hsu et al, 2022). Consistent with this, we identified an increased prevalence of schizophrenia in autistic New Zealanders. Previous studies have suggested there is a range of genetic variants at the intersection of these traits (Chen et al, 2021; Moreau et al, 2021; Rees et al, 2021). We identified 54 and

**Figure 2. Comparing traits identified during the multimorbid3D analysis for the fetal and adult groups.**
**(A)** Traits identified during the Multimorbid3D analysis shared between the adult and fetal groups using the STRING database to expand the protein–protein interaction network (PPIN). Some traits appear on multiple levels. Circle size indicates the number of expression quantitative trait loci (eQTLs), whereas colour indicates the negative-logged P-value from bootstrapping. Y-axis labels are based on GWAS trait names. Traits which passed the hypergeometric test ($P < 0.05$) but were insignificant ($P ≥ 0.05$) after bootstrapping are shaded grey. Note: neuroticism has also been misspelt as neuroticism in the GWAS Catalog meaning it appears twice. **(B)** Venn diagram outlining shared and unique traits across the autism PPIN (level 0–4). 44 traits appeared in both groups, representing 46% of the fetal traits and 52% of the adult traits. **(C)** Bar graph outlining the number of traits identified at different levels of the PPIN. The graph shows that most of the shared traits are at level 0 (**** = P-value < 0.0001 for shared traits and fetal traits. No difference in adult traits between index and outer levels; ns = not significant P-value = 0.179).

**Figure 3. De novo network analysis identified gene clusters and pleiotropic traits in the fetal gene regulatory networks for autism.**
Bi-clustering was performed on the gene trait associations using eQTL frequency to identify clusters. eQTL–gene data were obtained using Multimorbid3D. Brain and mood-related traits are shaded grey. Text coloured based on gene clusters. Black outlines indicate—the chromosome 17 genes (*KANSL1—RP11-259G18.1*) relating to a range of mood and brain traits, the chromosome 3 groups (*TMEM110—STAB1 and ITIH4—NT5DC2*) relating to cognition and worry, and the set of genes at the intersection of autism and schizophrenia. See Table S4 for raw output.

73 pleiotropic genes linking these phenotypes within the fetal and adult cortical tissues, respectively. These pleiotropic genes were distributed throughout the genome and the trait intersections were typically associated with a restricted number of eQTLs ($N_{fetal}$ = 2.6 ± 1.6, $N_{adult}$ = 2.3 ± 1.6). Notably, of the 50 eQTLs predicted by Sei to have the highest regulatory scores in the fetal and adult groups, 42% and 58% were associated with schizophrenia, respectively. In particular, *BORCS7*, *INA*, and *SRR* were associated with eQTLs which scored highly for reduced enhancer, promotor, and CTCF–cohesin activity. Polygenic risk score for schizophrenia has been shown to be predictive in autism (Antaki et al, 2022). However, here, we have identified the intersection, not simply the polygenic risk score but both the genetic risk and the possible mechanistic link. It also demonstrates that schizophrenia is linked to a subset of the possible genetic variation associated with autism, and not every gene. Collectively, these findings are consistent with a complex biological link that raises the chances of some autistic individuals exhibiting features of schizophrenia.

There was reduced clustering of traits—and genes—within the PPIN produced from the adult cortex, possibly reflecting the fact that autism is childhood onset. This relates to our finding of only one

causal autism gene (*LINC02210*) within the adult cortical tissue, compared with the four causal genes within the fetal tissue. Despite this, we identified a strong association with respiratory-related phenotypes, particularly on levels 0 (i.e., *CHRNA5*) and 1 (i.e., *CHRNA3*, *PSMA4*, *HYKK*). Of these, *PSMA4* was a pleiotropic gene within the fetal PPIN. *CHRNA5*, *CHRNA3*, *PSMA4*, and *HYKK* are located at 15q25.1, a locus that is associated with COPD (Nedeljkovic et al, 2018). Over 50 eQTLs were associated with *CHRNA5* which was associated with post-bronchodilator FEV1 and FEV1/FVC ratio. Kyoto Encyclopedia of Genes and Genomes (KEGG) pathway analysis also identified asthma as a pathway associated with the index and outer levels of the adult PPIN. Links between autism and lung-related phenotypes have rarely been documented, but one case study demonstrated that a group of 49 autistic children had abnormal lung anatomies when compared with 410 neurotypical control subjects (Stewart & Klar, 2013). However, no further validation of this study (Stewart & Klar, 2013) has occurred and studies on connections between autism and asthma show mixed results (Kotey et al, 2014; Zerbo et al, 2015; Gong et al, 2023). Despite this, SNPs associated with *CHRNA5* have been associated with a decrease in connectivity of a cortical output circuit linked with a cognitive profile that includes

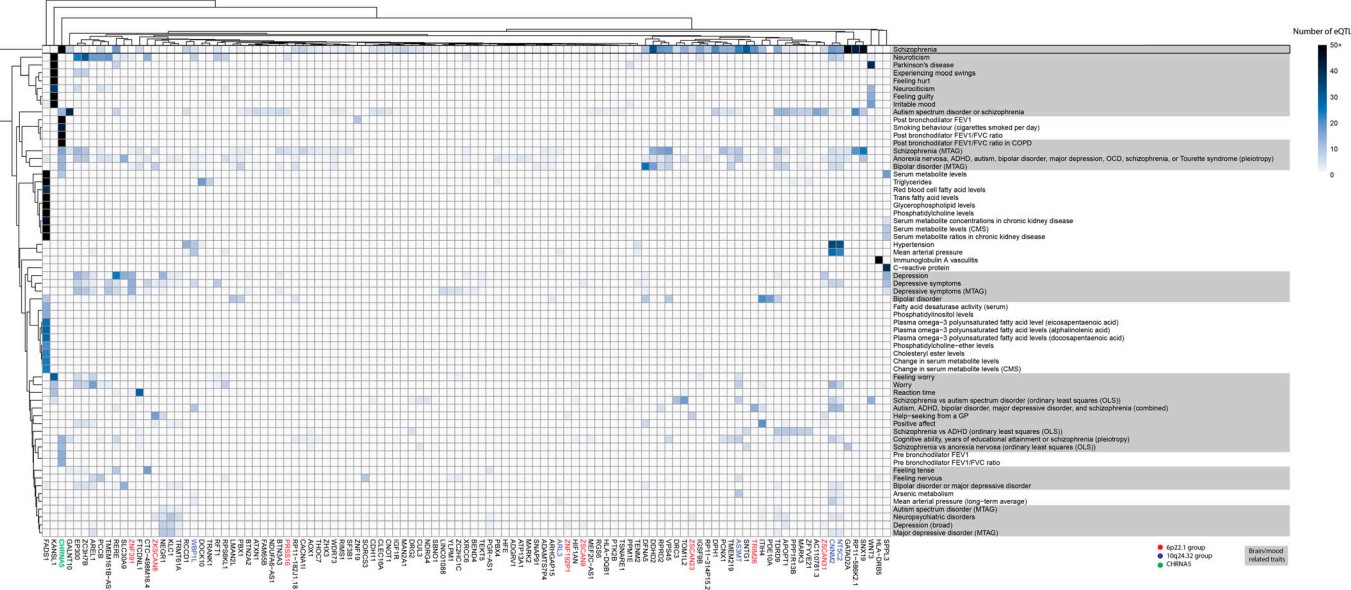

**Figure 4. De novo network analysis identified autism-associated gene clusters and pleiotropic traits within the adult cortex.**
Bi-clustering was performed on gene trait associations using eQTL numbers. Data were derived from a Multimorbid3D analysis of the adult cortical gene regulatory networks. For simplicity, all rows with only one eQTL–gene connection were removed (see Table S5 for raw output).

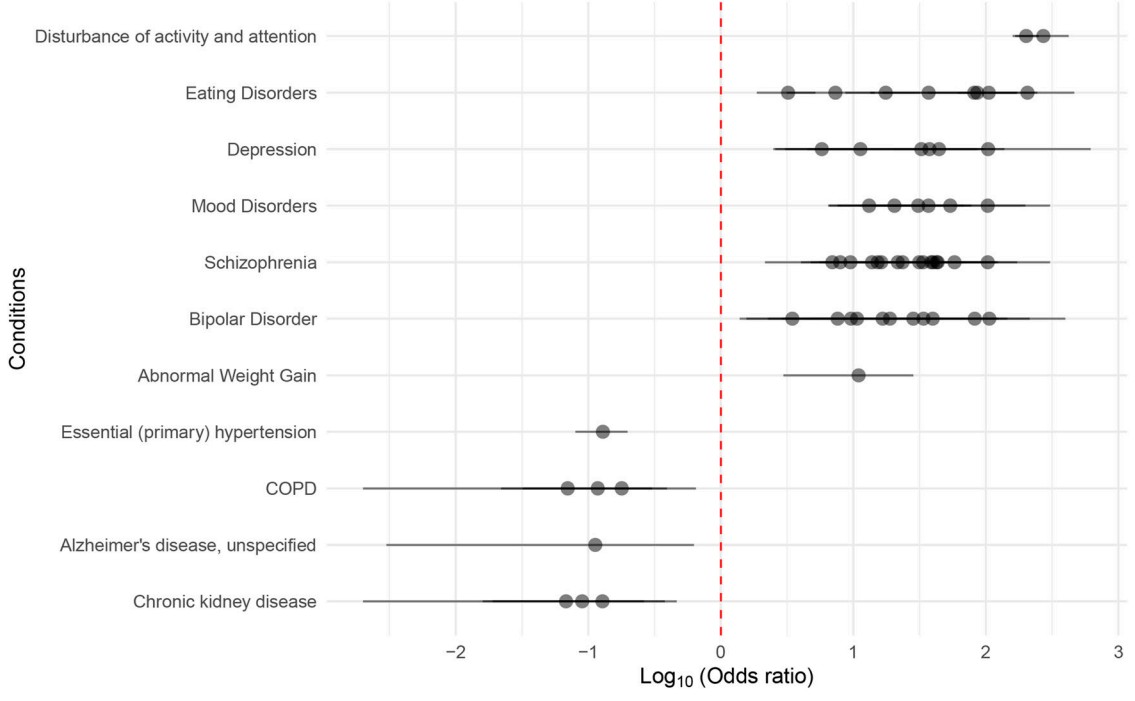

**Figure 5. Odds ratios for conditions identified as co-occurring within autistic New Zealanders.**
Conditions that occurred at significantly ($P < 0.05$ after multiple testing correction) different rates, when compared with the hospitalised New Zealand population, and overlapped traits identified by Multimorbid3D (i.e., Figs 3 and 4) are shown. Many conditions had multiple ICD-10 codes (e.g., childhood autism and Asperger's syndrome for autism and multiple subcodes for schizophrenia).

deficits in attention seen in autism (Bailey et al, 2012). We observed a decreased prevalence of COPD within autistic New Zealanders, suggesting autism is protective against COPD. Thus, although our study identifies a potential connection between autism and COPD, it is important to remember that gene regulation is tissue-specific, and our analysis was run on cortical tissue GRNs.

Parkinson's and Alzheimer's diseases were linked to autism through protein interactions (and thus, eQTL—gene associations) that appeared on the outer level of the fetal PPIN (Alzheimer's disease) and index and outer levels (Parkinson's disease) of both the fetal and adult PPINs. These neurodegenerative conditions were also identified as being enriched in the fetal pathway analysis, whereas "pathways of neurodegeneration—multiple diseases" was enriched in the adult analysis. Parkinson's disease was associated with the 17q21.31 loci within the fetal analysis, including *LINC02210* and *RP11-259G18.1* which were identified as causal autism genes. Parkinson's disease has been identified as being significantly more common in autistic individuals (Croen et al, 2015). Alzheimer's disease, Parkinson's disease, and autism have all been associated with neuronal nicotinic acetylcholine receptors (Henderson et al, 2012). Within the population of autistic New Zealanders, we saw a reduced prevalence of Alzheimer's disease and no statistically significant relationship between autism and Parkinson's disease. However, the analysis of autistic New Zealanders did not adjust for age and thus the apparent contradiction may be because of a smaller number of autistic individuals at an age where Parkinson's disease and Alzheimer's disease generally occur. Most standardised diagnostic testing for autism was not available until the 21st century, meaning many adults who would now display ageing diseases would have been unlikely to receive a diagnosis (Fombonne, 2009).

The findings from our network and epidemiological analyses provide insights into the time dependency of the genetic mechanisms involved in the interaction between autism and co-occurring traits. We have highlighted "developmental windows" (e.g., fetal development) where genetic loci (e.g., 17q21.31 and 3p21.1) have the potential to impact these trait combinations. This is supported by the identification of causal autism genes within the 17q21.31 region. It has previously been suggested that autism does not operate on a linear schedule, but is influenced by a combination of factors (Antaki et al, 2022). In this study, we provide a biological argument for looking at an individual's phenotype as being related to their combined genetic risk for different clusters of traits. Future work should test the utility of these genetic associations to create predictive models to stratify individuals for diagnosis and long-term management of the co-occurring conditions that confound autism and impact quality of life.

# Materials and Methods

## Creation of the fetal and adult cortical tissue GRNs

The CoDeS3D algorithm (Fadason et al, 2018) (https://github.com/Genome3d/codes3d) was used to develop GRNs for fetal and adult cortical tissues. All SNPs present in the adult cortical tissue-specific GTEx (GTEx Consortium, 2020) eQTL database (n = 46,569,705) and those from the fetal dataset (Walker et al, 2019) (n = 5,471,505) were input into CoDeS3D.

Hi-C chromatin libraries, derived from fetal brain-specific cortical plate and germinal zone neurons (Won et al, 2016) and adult dorsolateral prefrontal cortex cells (Schmitt et al, 2016),

were downloaded from dbGaP (accession: phs001190.v1.p1) and GEO (https://www.ncbi.nlm.nih.gov/geo/, accession: GSE87112), respectively.

This was used to generate a list of spatial SNP–gene pairs based on interactions between restriction fragments containing the queried SNPs and restriction fragments overlapping genes. From this, the adult (GTEx Consortium, 2020) and fetal (Walker et al, 2019) cortical tissue eQTL datasets were used to determine which SNPs were eQTLs (Fig 1A). Benjamini–Hochberg multiple testing was performed to identify significant (adjusted *P*-value ≤ 0.05) pairings.

## Two-sample Mendelian randomisation

To identify any genes potentially causal for autism within the adult and fetal cortical tissue GRNs, a 2SMR study was completed for adult and fetal separately (Fig S1A). This was done using the TwoSampleMR R package (https://github.com/MRCIEU/TwoSampleMR/, version 0.5.6) (Hemani et al, 2018). The eQTL–gene pairs within each adult and fetal cortical tissue GRN (output of Fig 1A) were used as the exposure instruments. All eQTLs with an exposure *P*-value > $1 \times 10^{-5}$ were also removed from the exposure dataset. Clumping was then undertaken to ensure all exposure instruments were independent (Hemani et al, 2018). The iPSYCH-PGS 2017 ASD GWAS (Grove et al, 2019) was chosen for the outcome data because of its size. This was downloaded within the TwoSampleMR package from the IEU Open GWAS Project (Elsworth et al, 2020 *Preprint*). After harmonising the exposure and outcome data, genes with one eQTL associated with it underwent 2SMR using the Wald test, whereas those with multiple eQTLs underwent 2SMR using MR Egger regression and inverse variance weighted methods (Dang et al, 2022). A Bonferroni correction was used to adjust the *P*-value threshold (0.05) to correct for multiple tests. Genes whose *P*-values were below this threshold were considered statistically significant 2SMR results and therefore have a putatively causal role in autism within adult or fetal cortical tissue.

## Definition of autism-associated SNPs

A keyword search for the exact term "Autism" identified 576 autism-associated SNPs ($P < 5 \times 10^{-8}$) in the GWAS Catalog (Table S1). These SNPs were downloaded (www.ebi.ac.uk/gwas; 08/05/2022).

## Identifying possible co-occurring traits of autism

Potential co-occurring traits were identified using the Multimorbid3D pipeline (Golovina et al, 2023; Zaied et al, 2023 *Preprint*) (https://github.com/Genome3d/multimorbid3D). The 576 autism-associated SNPs underwent a linkage disequilibrium analysis to determine those in LD ($r^2$ = 0.8, width = 5,000 bp; Table S2). 2,753 SNPs (n = 576 + LD) were input into Multimorbid3D which queried the fetal or adult cortical tissue-specific GRN created in Fig 1A to create an autism-specific fetal or adult GRN. Proteins encoded by the genes in this GRN formed the "Level 0" network. Protein interactions proximal to the autism-associated network (under four "edges" away; where an edge refers to a direct interaction) were identified using STRING (https://string-db.org; Fig 1B). Interactions came from experiments, text mining, co-expression, and databases,

and were limited to species "Homo sapiens" and an interaction score above 0.7 (Szklarczyk et al, 2019).

The "Level 1" proteins were those that were identified as interacting directly with the autism-associated proteins, whereas level 2–4 proteins were those interacting with proteins on the previous level. At each level, the genes encoding the proteins were used to query the fetal or adult cortical tissue GRNs to identify their associated significant regulatory eQTLs (adjusted *P*-value ≤ 0.05). These eQTLs at each level were queried against the GWAS Catalog to identify traits associated with them (hypergeometric test; *P* < 0.05). This process was completed for both fetal and adult cortical tissue separately to create the fetal and adult PPINs.

### Bootstrapping analysis

After the Multimorbid3D analysis, 1,000 bootstrapping iterations were performed to mitigate any errors from the traditional sampling approach assumption that the sampling distribution will be approximately normal. At each iteration, 2,753 trait-associated SNPs were randomly selected from the GWAS Catalog and run through the Multimorbid3D pipeline. After 1,000 iterations, the *P*-value for each trait was calculated by counting the instances in which the number of SNPs in the bootstrapped overlap was greater than or equal to the number of SNPs in the observed overlap.

$$Pvalue_{bootstrap} = \frac{\sum(bootrapped \geq observed)}{1000}$$

If *Pvalue*$_{bootstrap}$ < 0.01, we assume that the observed relationship for that SNP is not random.

### Pathway analysis

Pathway enrichment analysis was completed using g:profiler and the Kyoto Encyclopedia of Genes and Genomes (Kanehisa et al, 2019), through the gprofiler2 R package. False discovery rate multiple testing was used to select pathways with an adjusted *P*-value < 0.05 (Table S6).

### Using deep learning to identify regulatory activity of SNPs associated with autism

To estimate the regulatory activity of SNPs associated with autism, the 2,753 SNPs (autism-associated GWAS SNPs and those in LD) were queried into Sei, a deep learning algorithm for functionally annotating SNPs (Chen et al, 2022). Sei scored every SNP based on their predicted regulatory activity across 40 groups (Chen et al, 2022) (Table S7). The SNPs were also separated into those that were spatially constrained adult and/or fetal eQTLs from those that were not (Fig 1C). These groups were then compared based on their predicted regulatory activity and mean scores.

### Identification of traits that co-occur with autism within the New Zealand population

For robustness, we compared the traits we determined to co-occur with autism with those seen in the New Zealand population. The Integrated Data Infrastructure is a database maintained by Stats NZ that contains de-identified microdata about individuals in New Zealand (Statistics New Zealand, 2022). The comoRbidity R package (Gutiérrez-Sacristán et al, 2018) was used to identify conditions that co-occur with autism within New Zealand's hospital admissions data (31 December 2015 –1 January 2021). Conditions were based on the ICD-10 codes with F840 (childhood autism) and F845 (Asperger's syndrome) used for autism. Within the comoRbidity package, Fisher's exact tests were performed to calculate odds ratios for F840 and F845 occurring with all other ICD-10 codes. Odds ratios which were significant (corrected *P*-value after multiple testing ≤ 0.05) were selected.

### Ethics statement

Use of the Integrated Data Initiative within Stats NZ Data Laboratory was reviewed and approved by Statistics New Zealand (project number MAA2020-63) and the Auckland Health Research Ethics Committee (approval AH22495).

## Data Availability

Data analyses and visualisations were performed in R (version 4.2.0) through RStudio (version 2022.02.2). The Multimorbid3D pipeline, including bootstrapping, was performed in Python (version 3.8.8). Sei analysis was also performed on Python, using Jupyter Notebooks (version 6.3.0). Datasets and software used in the analysis are listed in Table S9. All scripts are available on github (https://github.com/Catriona-Miller/Autism_Co-occurring_Traits).

## Supplementary Information

## Acknowledgements

We would like to thank the Genomics and Systems Biology Group (Liggins Institute, University of Auckland) for their helpful suggestions and discussions. Data from this work come from the Genotype-Tissue Expression (GTEx) Project, which was supported by the Common Fund of the Office of the Director of the National Institutes of Health, and by NCI, NHGRI, NHLBI, NIDA, NIMH, and NINDS. CJ Miller was funded by the University of Auckland Doctoral Scholarship. E Golovina and JM O'Sullivan were funded by the Dines Family Foundation.

### Author Contributions

CJ Miller: formal analysis, validation, visualization, methodology, and writing—original draft, review, and editing.
E Golovina: supervision, methodology, and writing—review and editing.
JS Wicker: supervision, methodology, and writing—review and editing.

JC Jacobsen: supervision, methodology, and writing—review and editing.

JM O'Sullivan: conceptualization, supervision, funding acquisition, methodology, project administration, and writing—review and editing.

## Conflict of Interest Statement

The authors declare that they have no conflict of interest.

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
