## [Reviewer comments · Life Science Alliance]

Life Science Alliance

De novo network analysis reveals autism causal genes and developmental links to co-occurring traits.

Catriona Miller, Evgeniia Golovina, Joerg Wicker, Jessie Jacobsen, and Justin O'Sullivan

DOI: <https://doi.org/10.26508/lsa.202302142>

Corresponding author(s): *Justin O'Sullivan, The University of Auckland*

Review Timeline:

Submission Date:	2023-05-08
Editorial Decision:	2023-07-14
Revision Received:	2023-07-21
Editorial Decision:	2023-07-25
Revision Received:	2023-07-26
Accepted:	2023-07-27

Scientific Editor: Novella Guidi

Transaction Report:

July 14, 2023

Re: Life Science Alliance manuscript #LSA-2023-02142-T

Dr. Justin M. O'Sullivan
The University of Auckland
The Liggins Institute
University of Auckland
Private Bag 92019
Auckland 1142
New Zealand

Dear Dr. O'Sullivan,

Thank you for submitting your manuscript entitled "An unbiased de novo network analysis uncovering causal genes and the developmental intersection between autism and co-occurring traits" to Life Science Alliance. The manuscript was assessed by expert reviewers, whose comments are appended to this letter. We invite you to submit a revised manuscript addressing the Reviewer comments.

Thank you for this interesting contribution to Life Science Alliance. We are looking forward to receiving your revised manuscript.

Sincerely,

B. MANUSCRIPT ORGANIZATION AND FORMATTING:

Reviewer #1 (Comments to the Authors (Required)):

Thank you for this opportunity to review this manuscript.

I consider the work presented here to be excellent. From my perspective the appropriate tools have been used to examine an important question. But more than that, the approach modeled here is crucially important to understanding the basis of complex diseases and is an important step away from a single-gene causative interpretation.

I have only minor issues to raise:

1. I am slightly uncomfortable with the term 'causative' as cause has not been proven in this work.
2. the introduction is quite dense, does not emphasize that this work is using human data sets
3. ADHD needs to be defined
4. MAPT encodes tau

Reviewer #2 (Comments to the Authors (Required)):

Autism is a complex neurodevelopmental condition that manifests in various ways. Autism is often accompanied by other neurological disorders, the role of specific genes in autism, their relationship with co-occurring traits is not fully understood, the author of this paper conducted a two-sample Mendelian Randomisation analysis and identified four genes located at the 17q21.31 locus that are causally linked to autism in fetal cortical tissue (i.e. LINC02210, LRR37A4P, RP11-259G18.1, RP11-798G7.6). LINC02210 was also identified as being causally related to autism in adult cortical tissue, their results support that an individual's autism phenotype is partially determined by their genetic risk for co-occurring conditions, which could be used to develop predictive models for more accurate diagnosis and better clinical management. However, there are some concerns that are not well addressed, therefore, I would strongly suggest the author provide a major revision before acceptance, the comments are as follows:

Major concern:

1. Although the author provided four genes located at the 17q21.31 locus that are causally linked to autism in fetal cortical tissue (i.e. LINC02210, LRR37A4P, RP11-259G18.1, RP11-798G7.6), however, the author failed to verify their function, at least expression measurement.
2. Based on data of this paper, we do not know whether these identified genes are neuronal or nonneuronal expression, inhibitory or excitatory neurons expression.
3. How many samples were used to generate these many data, although we can see that these data were extracted from somewhere, but I think you should provide more detail regarding these samples being used.
4. Are there any differences of these identified genes expression in different gender or sex.
5. Did you check these identified gene expression level in PD, AD and autism, since the author found that LINC02210 and RP11-259G18.1 were associated with PD, AD and autism.
6. Did you evaluate the prevalence of schizophrenia in autistic other populations?

Minor concern:

1. Please provide full name of abbreviated words, such as ADHD, GTEX et.

Dear Sir/Madam

We would like to thank you and the reviewers for the opportunity to modify our manuscript following the constructive criticisms that were raised by the referees. Both referees were positive about our work and their comments have helped us make changes that have improved the quality of the manuscript. We have provided a point-by-point response to the referees below.

Sincerely

Justin M. O'Sullivan

Reviewer 1:

1. I am slightly uncomfortable with the term 'causative' as cause has not been proven in this work.
 - We have updated the text to indicate that Mendelian Randomisation identifies genes that are putatively causal:
 - *"After 2SMR, four genes (LINC02210, LRRC37A4P, RP11-259G18.1, RP11-798G7.6) were identified as being statistically significant (Bonferroni adjusted p-value < 0.05) and having a putatively causal role in autism (here on referred to as causal) within fetal cortical tissue (supplementary figure 1b)."*
2. The introduction is quite dense, does not emphasize that this work is using human data sets
 - We have shortened the introduction and updated the text to emphasise the use of human datasets:
 - *"To investigate the interconnectivity of complex polygenic traits, network-based analyses using human derived datasets s have been previously used by our group (Golovina et al. 2023). In this study, we identified causative and pleiotropic genes that are affected by SNPs associated with both autism and other traits within human cortical gene regulatory networks (GRNs) at two different developmental stages."*
3. ADHD needs to be defined
 - We have defined ADHD at first mention:
 - *"Common co-occurring conditions reported with autism include attention-deficit/hyperactivity disorder (ADHD), anxiety, depressive disorders and schizophrenia."*
4. MAPT encodes tau

- The text has been corrected to state that MAPT encodes tau:

“For example, MAPT encodes tau, potentially resulting in the development of tauopathy in the brain (Grigg et al. 2020).”

Reviewer 2:

1. Although the author provided four genes located at the 17q21.31 locus that are causally linked to autism in fetal cortical tissue (i.e. LINC02210, LRRC37A4P, RP11-259G18.1, RP11-798G7.6), however, the author failed to verify their function, at least expression measurement.

- The referee is correct, we were unable to verify the function of the four genes located in 17q21.31. Firstly, these genes are non-coding/pseudogenes and we have been unable to extrapolate on their function. We were unable to make direct measures of the expression of these genes in Autistic tissues. However, we modified the manuscript to clarify that these ncRNAs/pseudogenes are expressed in the fetal and adult cortical tissues:

“After 2SMR, four genes (LINC02210, LRRC37A4P, RP11-259G18.1, RP11-798G7.6) were identified as being statistically significant (Bonferroni adjusted p-value < 0.05) and having a putatively causal role in autism (here on referred to as causal) within fetal cortical tissue (supplementary figure 1b). These four genes are expressed in the fetal cortical tissue GRN with gene expression values of 17.38 (LINC02210), 8.99 (LRRC37A4P), 0.88 (RP11-259G18.1), and 1.35 (RP11-798G7.6) TPM. Within the adult cortical tissue GRN, only LINC02210 was identified as being statistically significant. It is expressed in the adult cortical tissue GRN (3.97 TPM).”

“As the causal genes identified are all non-coding or pseudogenes, little is known about their function. Future work should explore the role these four genes play in the development of autism.”

2. Based on data of this paper, we do not know whether these identified genes are neuronal or nonneuronal expression, inhibitory or excitatory neurons expression.
 - We regret we have been unable to clarify this. See reply to Reviewer 2 point 1.
3. How many samples were used to generate these many data, although we can see that these data were extracted from somewhere, but I think you should provide more detail regarding these samples being used.
 - The data used in this study were obtained from multiple sources (see supplementary table 9 for data accession codes). Information on Hi-C and GTEx sample numbers is best obtained from the relevant references which are provided in the text (GTEx Consortium, 2020; Schmitt et al., 2016; Walker et al., 2019; Won et al., 2016). We have modified supplementary figure 1a to provide the sample details for the GWAS study used for 2SMR.

4. Are there any differences of these identified genes expression in different gender or sex.
 - We thank the reviewer for their suggestion. This is something we often discuss, however, we have not yet investigated it largely due to limitations with our current datasets. We hope to be able to address this with appropriate datasets in a future study.
5. Did you check these identified gene expression level in PD, AD and autism, since the author found that LINC02210 and RP11-259G18.1 were associated with PD, AD and autism.
 - We have information in independent studies that indicate there is some convergence on causal genes in the 17q21.31 locus in PD. This is currently in preparation. We regret that we have not yet analyzed AD.
6. Did you evaluate the prevalence of schizophrenia in autistic other populations?
 - We have not investigated the prevalence of schizophrenia in autistic other populations. This was outside the scope of this study.
7. Please provide full name of abbreviated words, such as ADHD, GTEx et.

- We have defined all abbreviations (including ADHD and GTEx) at the first mention.

References:

- Grigg, I., Ivashko-Pachima, Y., Hait, T. A., Korenková, V., Touloumi, O., Lagoudaki, R., van Dijck, A., Marusic, Z., Anicic, M., Vukovic, J., Kooy, R. F., Grigoriadis, N., & Gozes, I. (2020). Tauopathy in the young autistic brain: novel biomarker and therapeutic target. *Translational Psychiatry, 10*(1). <https://doi.org/10.1038/s41398-020-00904-4>
- GTEx Consortium. (2020). The GTEx Consortium atlas of genetic regulatory effects across human tissues. *Science, 1318*–1330. <https://doi.org/10.5281/zenodo.3727189>
- Schmitt, A. D., Hu, M., Jung, I., Xu, Z., Qiu, Y., Tan, C. L., Li, Y., Lin, S., Lin, Y., Barr, C. L., & Ren, B. (2016). A Compendium of Chromatin Contact Maps Reveals Spatially Active Regions in the Human Genome. *Cell Reports, 17*(8), 2042–2059. <https://doi.org/10.1016/j.celrep.2016.10.061>
- Walker, R. L., Ramaswami, G., Hartl, C., Mancuso, N., Gandal, M. J., de la Torre-Ubieta, L., Pasaniuc, B., Stein, J. L., & Geschwind, D. H. (2019). Genetic Control of Expression and Splicing in Developing Human Brain Informs Disease Mechanisms. *Cell, 179*(3), 750-771.e22. <https://doi.org/10.1016/j.cell.2019.09.021>
- Won, H., De La Torre-Ubieta, L., Stein, J. L., Parikshak, N. N., Huang, J., Opland, C. K., Gandal, M. J., Sutton, G. J., Hormozdiari, F., Lu, D., Lee, C., Eskin, E., Voineagu, I., Ernst, J., & Geschwind, D. H. (2016). Chromosome conformation elucidates regulatory relationships in developing human brain. *Nature, 538*(7626), 523–527. <https://doi.org/10.1038/nature19847>

July 25, 2023

RE: Life Science Alliance Manuscript #LSA-2023-02142-TR

Dr. Justin M. O'Sullivan
The University of Auckland
The Liggins Institute
University of Auckland
Private Bag 92019
Auckland 1142
New Zealand

Dear Dr. O'Sullivan,

Thank you for submitting your revised manuscript entitled "De novo network analysis reveals autism causal genes and developmental links to co-occurring traits.". We would be happy to publish your paper in Life Science Alliance pending final revisions necessary to meet our formatting guidelines.

- please consult our manuscript preparation guidelines <https://www.life-science-alliance.org/manuscript-prep> and make sure your manuscript sections are in the correct order
- all figure legends should only appear in the main manuscript file
- please add your main, supplementary figure, and table legends to the main manuscript text after the References section
- please add a Category for your manuscript in our system
- the Supplemental Tables Index can be removed

A. FINAL FILES:

B. MANUSCRIPT ORGANIZATION AND FORMATTING:

Sincerely,

July 27, 2023

RE: Life Science Alliance Manuscript #LSA-2023-02142-TRR

Dr. Justin M. O'Sullivan
The Liggins Institute
University of Auckland
Private Bag 92019
Auckland 1142
New Zealand

Dear Dr. O'Sullivan,

Thank you for submitting your Research Article entitled "De novo network analysis reveals autism causal genes and developmental links to co-occurring traits.". It is a pleasure to let you know that your manuscript is now accepted for publication in Life Science Alliance. Congratulations on this interesting work.

DISTRIBUTION OF MATERIALS:

Again, congratulations on a very nice paper. I hope you found the review process to be constructive and are pleased with how the manuscript was handled editorially. We look forward to future exciting submissions from your lab.

Sincerely,
